# Nucleosome breathing and remodeling constrain CRISPR-Cas9 function

R Stefan Isaac[1,2], Fuguo Jiang[3,4], Jennifer A Doudna[4,5,6,7,8], Wendell A Lim[9,10,11]*, Geeta J Narlikar[1]*, Ricardo Almeida[10,11,12]

[1]Department of Biochemistry and Biophysics, University of California, San Francisco, San Francisco, United States; [2]Tetrad Graduate Program, University of California, San Francisco, San Francisco, United States; [3]Department of Molecular and Cell Biology, University of California, Berkeley, Berkeley, United States; [4]California Institute for Quantitative Biosciences, University of California, Berkeley, Berkeley, United States; [5]Department of Molecular and Cell Biology, Howard Hughes Medical Institute, University of California, Berkeley, Berkeley, United States; [6]Department of Chemistry, University of California, Berkeley, Berkeley, United States; [7]Physical Biosciences Division, Lawrence Berkeley National Laboratory, Berkeley, United States; [8]Innovative Genomics Initiative, University of California, Berkeley, Berkeley, United States; [9]Department of Cellular and Molecular Pharmacology, Howard Hughes Medical Institute, University of California, San Francisco, San Francisco, United States; [10]Center for Systems and Synthetic Biology, University of California, San Francisco, San Francisco, United States; [11]California Institute for Quantitative Biosciences, University of California, San Francisco, San Francisco, United States; [12]Department of Cellular and Molecular Pharmacology, University of California, San Francisco, San Francisco, United States

*For correspondence: Wendell. Lim@ucsf.edu (WAL); Geeta. Narlikar@ucsf.edu (GJN)

**Abstract** The CRISPR-Cas9 bacterial surveillance system has become a versatile tool for genome editing and gene regulation in eukaryotic cells, yet how CRISPR-Cas9 contends with the barriers presented by eukaryotic chromatin is poorly understood. Here we investigate how the smallest unit of chromatin, a nucleosome, constrains the activity of the CRISPR-Cas9 system. We find that nucleosomes assembled on native DNA sequences are permissive to Cas9 action. However, the accessibility of nucleosomal DNA to Cas9 is variable over several orders of magnitude depending on dynamic properties of the DNA sequence and the distance of the PAM site from the nucleosome dyad. We further find that chromatin remodeling enzymes stimulate Cas9 activity on nucleosomal templates. Our findings imply that the spontaneous breathing of nucleosomal DNA together with the action of chromatin remodelers allow Cas9 to effectively act on chromatin *in vivo*.

## Introduction

The recent development of CRISPR (*c*lustered *r*egularly *i*nterspaced *s*hort *p*alindromic *r*epeats) systems, particularly the type II CRISPR-Cas9 mechanism from *Streptomyces pyogenes*, as an artificial tool for genome engineering, gene regulation, and live imaging is a remarkable achievement with profound impact in a wide variety of research fields and applications (*Makarova et al., 2015*; *Doudna and Charpentier, 2014*; *Cong et al., 2013*; *Jinek et al., 2012*; *2013*; *Mali et al., 2013*). Despite its successful adoption across numerous eukaryotic organisms, relatively few details are known of the mechanism by which bacterial CRISPR-Cas9 systems operate in eukaryotic cells (*Doudna and Charpentier, 2014*; *Ghorbal et al., 2014*; *Vyas et al., 2015*).

**eLife digest** CRISPR is a method of editing the genetic material inside living cells and has enabled dramatic advances in a broad variety of research fields in recent years. The method relies on a bacterial enzyme called Cas9 that can be programmed, via short guide molecules made from RNA, to target specific sites in the cell's DNA. Once bound to its target, the Cas9 enzyme cuts the DNA molecule; this often leads to changes in the DNA sequence. In nature, bacteria use the CRISPR-Cas9 system to defend themselves against viruses. However, this system also works in other cell types and can be reprogrammed to target almost any site in the DNA.

To date, the CRISPR-Cas9 system has been used in fungi, worms, flies, plants, mammals and other eukaryotes. Yet, unlike in bacteria, much of the DNA in eukaryotes is wrapped around proteins called histones to form units referred to as nucleosomes. This means eukaryotic DNA is often tightly packaged, which makes it less accessible to other proteins. Nevertheless, eukaryotic DNA will spontaneously detach and reattach to the histones – a phenomenon that is commonly known as DNA "breathing". Also, protein machines known as chromatin remodelers can move, assemble and take apart the nucleosomes in eukaryotic cells. However, because there is much still to learn about how CRISPR-Cas9 works in eukaryotic cells, it is not clear how nucleosomes affect this system's activity.

Isaac et al. have now used a simplified biochemical system to test how nucleosomes and chromatin remodelers affect CRISP-Cas9 activity. The system comprised purified Cas9 enzymes, short guide RNA molecules and nucleosomes. The experiments revealed that the Cas9 enzyme was able to cut DNA on nucleosomes when the DNA sequence allowed more spontaneous breathing or when chromatin remodelers were present to destabilize or move the nucleosome out of the way.

These results suggest that by taking the placement of the nucleosomes into account, researchers can better predict how effective the CRISPR-Cas9 system will be at targeting a specific DNA sequence in a eukaryotic cell. The findings also suggest ways to make genome editing with CRISPR-Cas9 even more efficient.

CRISPR-Cas9 originated in bacteria, where genomic DNA generally consists of supercoiled circular molecules associated with nucleoid-associated proteins (*Travers and Muskhelishvili, 2005*). In contrast, eukaryotic chromosomes are linear, packaged with histone octamers into nucleosomes, and further organized into higher-order structures (*Luger et al., 1997*; *Olins and Olins, 1974*; *Woodcock et al., 1976*; *Dixon et al., 2012*). The packaging of DNA into nucleosomes generally inhibits the binding of sequence specific DNA binding factors. In the simplest model, nucleosomes would analogously inhibit Cas9 action. Further, in eukaryotes ATP-dependent chromatin remodelers reposition, remove, or restructure nucleosomes to regulate the access of DNA binding factors (*Clapier and Cairns, 2009*; *Narlikar et al., 2013*). It can therefore be imagined that the action of remodelers also regulates the action of Cas9 on nucleosomes.

To quantitatively test the above models we performed biochemical studies to measure Cas9 activity on nucleosomes assembled with native and artificial nucleosome positioning sequences. We find that the combination of nucleosome breathing, by which DNA transiently disengages from the histone octamer, and the action of chromatin remodeling enzymes allow Cas9 to act on nucleosomal DNA with rates comparable to naked DNA. The results provide a biochemical explanation for the efficacy of Cas9 in eukaryotic cells.

## Results

### Nucleosomes assembled on the 601 sequence inhibit Cas9 binding and cleavage of target DNA

To determine if a nucleosome inhibits the ability of Cas9 to scan, recognize, and cleave sgRNA-directed DNA targets, we performed *in vitro* Cas9 cleavage assays using mononucleosomes (single nucleosomes on short dsDNA molecules) reconstituted using the Widom 601 positioning sequence with 80 base pairs of flanking DNA on both sides (referred to as 601 80/80 particles, *Figure 1A*)

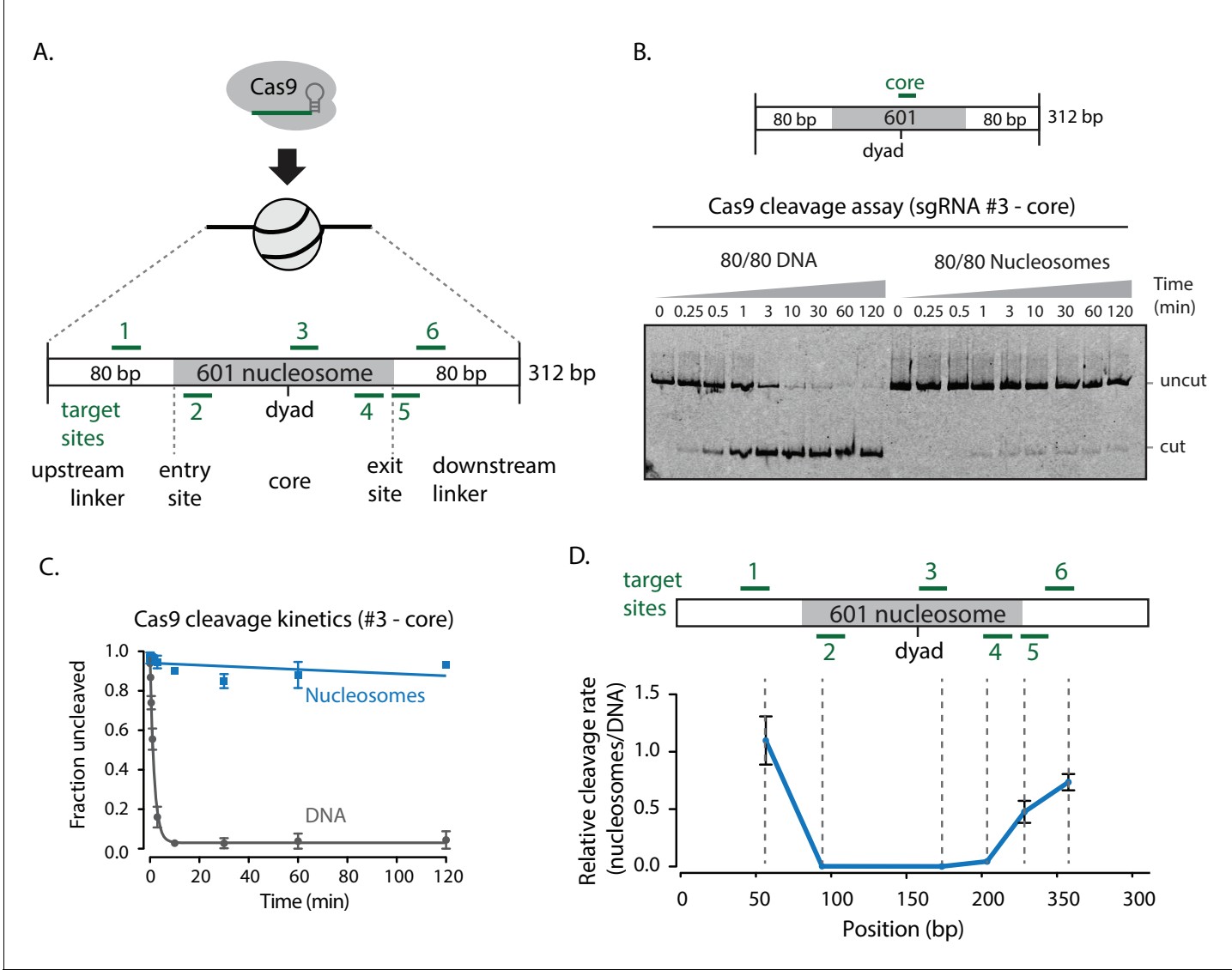

**Figure 1.** Cas9 DNA nuclease activity is hindered by nucleosomes. (**A**) Schematic of sgRNAs designed against the assembled 601 80/80 nucleosome substrates targeting the flanking regions, entry/exit sites, and near the nucleosomal dyad. (**B**) Cleavage assay comparing Cas9 cleavage on 80/80 DNA and 80/80 nucleosomes when loaded with sgRNA #3. (**C**) Kinetics of cleavage with sgRNA #3. (**D**) Comparison of the relative rates of cleavage on nucleosomes to DNA at various positions along the 80/80 nucleosome construct. The position reported is the site of cleavage by Cas9. Represented values are mean ± SEM from three replicates.

The following source data and figure supplements are available for figure 1:

**Source data 1.** Replicate gels of Cas9 cleavage of 80/80 601 DNA and nucleosomes with sgRNAs #2 and #6.

**Source data 2.** Replicate gels of Cas9 cleavage of 80/80 601 DNA and nucleosomes with sgRNAs #2 and #6.

**Source data 3.** Replicate gels of Cas9 cleavage of 80/80 601 DNA and nucleosomes with sgRNAs #2 and #6.

**Source data 4.** Replicate gels of Cas9 cleavage of 80/80 601 DNA and nucleosomes with sgRNA #5.

**Source data 5.** Replicate gels of Cas9 cleavage of 80/80 601 DNA and nucleosomes with sgRNA #5.

**Source data 6.** Replicate gels of Cas9 cleavage of 80/80 601 DNA and nucleosomes with sgRNA #1.

**Source data 7.** Replicate gels of Cas9 cleavage of 80/80 601 DNA and nucleosomes with sgRNA #1.

*Figure 1 continued on next page*

*Figure 1 continued*

**Source data 8.** Replicate gels of Cas9 cleavage of 80/80 601 DNA and nucleosomes with sgRNA #3.

**Source data 9.** Replicate gels of Cas9 cleavage of 80/80 601 DNA and nucleosomes with sgRNA #3.

**Source data 10.** Replicate gels of Cas9 cleavage of 80/80 601 DNA and nucleosomes with sgRNA #4.

**Source data 11.** Replicate gels of Cas9 cleavage of 80/80 601 DNA and nucleosomes with sgRNA #4.

**Source data 12.** Quantification of *Figure 1* Cas9 cleavage gels.

**Figure supplement 1.** Nucleosome positioning blocks Cas9 from binding PAM sites on DNA.

**Figure supplement 1—source data 1.** -3Replicate gels of dCas9 binding to 0/0 601 DNA and nucleosomes with sgRNA #3.

**Figure supplement 1—source data 2.** -3Replicate gels of dCas9 binding to 0/0 601 DNA and nucleosomes with sgRNA #3.

**Figure supplement 1—source data 3.** -3Replicate gels of dCas9 binding to 0/0 601 DNA and nucleosomes with sgRNA #3.

**Figure supplement 1—source data 4.** Quantification of *Figure 1—figure supplement 1* gel shifts.

(*Lowary and Widom, 1998*). The 601 sequence is an artificially derived sequence with high affinity for the histone octamer and has proved a valuable tool for assembling well positioning nucleosomes for biochemical studies. Using sgRNAs targeting the nucleosomal dyad, entry/exit sites, and flanking DNA, we measured the rates of Cas9 cleavage with naked 601 DNA and the 601 80/80 particles. Targeting the DNA flanking the nucleosome showed cleavage rates comparable to those of naked DNA. Cleavage rates at entry/exit sites of the nucleosome were much lower compared to naked DNA (∼23–28x decrease cleavage rate vs. DNA alone) (*Figure 1B,C*). Targeting near the nucleosomal dyad resulted in further inhibition of cutting by Cas9 (∼1000x decrease vs. DNA alone) (*Figure 1C,D*). Previous work has shown that nucleosomal DNA transiently disengages from the histone octamer, a process termed as nucleosomal DNA unpeeling or breathing. The equilibrium for DNA unpeeling gets progressively more unfavorable the closer the DNA site gets to the dyad (*Polach and Widom, 1995*; *Li and Widom, 2004*; *Luger et al., 2012*). The nucleosome-mediated inhibition of Cas9 activity is more pronounced near the dyad suggesting that Cas9 cleavage occurs on DNA that is transiently disengaged from the histone octamer.

Nucleosomes block the ability of Cas9 to cleave DNA, but it is unclear at which step of Cas9 activity this occurs. Cas9 recognizes DNA target sites in a two-step process that begins with binding to the DNA protospacer adjacent motif (PAM, in this case 'NGG') through its C-terminal PAM-interacting region, followed by sequential melting of the DNA double strand and annealing of the sgRNA guide segment to the unwound target DNA strand (*Figure 1—figure supplement 1A*) (*Sternberg et al., 2014*; *Jiang et al., 2015*). Complete annealing of the 20-nt guide RNA to the target DNA is required to drive a progressive conformational transformation that authorizes Cas9 to simultaneously cleave both DNA strands (*Sternberg et al., 2015*; *Josephs et al., 2016*). Given this order of events, it is conceivable that nucleosomes can interfere with any of the steps preceding and including DNA cleavage.

To identify the point at which nucleosomes disrupt Cas9 function, we assessed binding of nuclease-dead Cas9 (dCas9) to mononucleosomal particles by an electrophoretic mobility shift assay. We performed dCas9 binding assays using 601 0/0 nucleosomal particles which are devoid of naked DNA overhangs. Binding of dCas9 pre-loaded with core targeting sgRNA with 601 0/0 nucleosomes is undetectable whereas binding to naked DNA control is still robust (*Figure 1—figure supplement 1B*). The presence of super shifts in the gel migration pattern suggests that multiple dCas9 molcules are engaging the same DNA substrate molecule. We investigated this further and determined that, in our binding assay, the highly transient dCas9 binding to PAMs within target DNA is observable as super shifts, likely due to a combination of a high number of PAMs on the target DNA (23 NGG

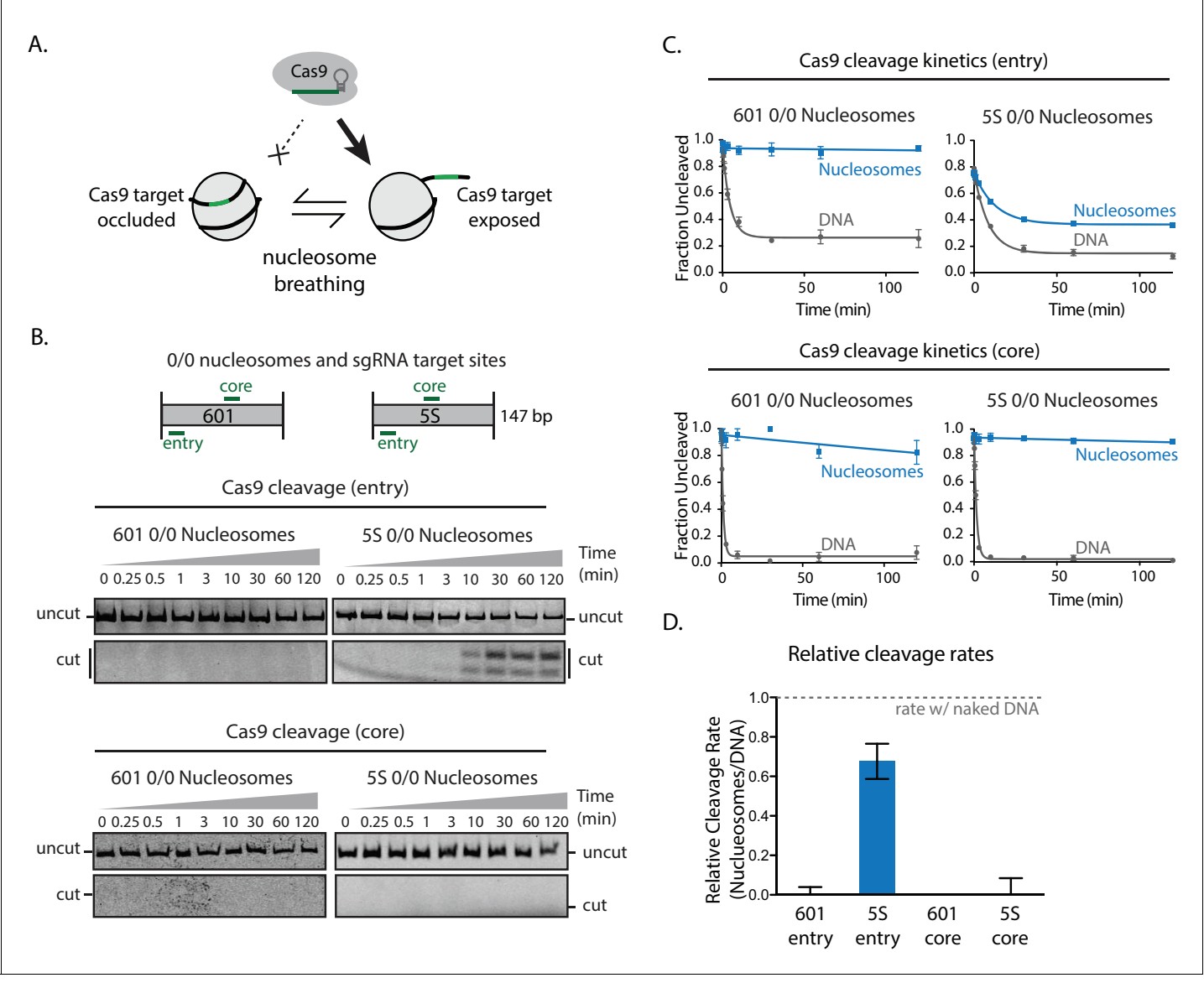

**Figure 2.** Higher nucleosomal breathing dynamics enhance Cas9 cleavage. (**A**) Schematic illustrating nucleosome breathing and how it can enable Cas9 binding to a target in the nucleosome. (**B**) Cleavage assay comparing Cas9 cleavage of 601 and 5S 0/0 nucleosomes when loaded with sgRNAs targeting comparable positions at core and entry sites. (**C**) Quantification of (**B**). (**D**) Cas9 cleavage rates on 601 and 5S nucleosomes when targeted to core and entry sites. Values were normalized against naked DNA control rates. Represented values are mean ± SEM from three replicates. Additional gel panels shown in *Figure 2—figure supplement 1*.

The following source data and figure supplement are available for figure 2:

**Source data 1.** Replicate gels of cleavage of 0/0 5S DNA and nucleosomes with sgRNA core.

**Source data 2.** Replicate gels of cleavage of 0/0 5S DNA and nucleosomes with sgRNA core.

**Source data 3.** Replicate gels of cleavage of 0/0 5S DNA and nucleosomes with sgRNA entry.

**Source data 4.** Replicate gels of cleavage of 0/0 5S DNA and nucleosomes with sgRNA entry.

**Source data 5.** Replicate gels of cleavage of 0/0 601 DNA and nucleosomes with sgRNA entry.

**Source data 6.** Replicate gels of cleavage of 0/0 601 DNA and nucleosomes with sgRNA entry.

*Figure 2 continued on next page*

*Figure 2 continued*

**Source data 7.** Quantification of *Figure 2* Cas9 cleavage gels.

**Source data 8.** Quantification of *Figure 2* Cas9 cleavage gels.

**Figure supplement 1.** Cas9 cleavage assay with 601 and 5S 0/0 nucleosomes.

PAMs present in 601 0/0 sequence) and common caging effects of gel binding assays. The absence of a super shift binding pattern with 0/0 nucleosomes (*Figure 1—figure supplement 1B*, right) suggests that dCas9 cannot stably interact with PAMs located on nucleosomes, in a manner consistent with a recently published study (*Hinz et al., 2015*).

## Nucleosomes assembled on a native DNA sequence are permissive to Cas9 action

The artificial Widom 601 is an atypically strong nucleosome positioning sequence that shows ~100-fold less breathing dynamics compared to physiological nucleosome positioning sequences, such as the 5S rRNA gene (*Anderson et al., 2002*; *Partensky and Narlikar, 2009*). To determine how Cas9 contends with nucleosomes assembled on this physiological positioning sequence, we performed cleavage experiments with nucleosomes assembled from 5S rRNA gene sequences from *Xenopus borealis* (*Figure 2A*). Cas9-mediated cleavage at sites near the entry/exit of the nucleosome is substantially enhanced (700–fold) with 5S nucleosomes compared to 601 particles (*Figure 2B–D*). In the context of 601, cutting at this site is 1000-fold slower than in naked DNA. In contrast, with 5S nucleosomes, cutting at the comparable site is only 1.5-fold slower than in naked DNA. However, Cas9 cleavage near the dyad is inhibited to a similar extent on both 5S and 601 nucleosomes, showing that the 5S-specific enhancement of Cas9 activity does not extend all the way to the nucleosomal dyad. These results support our interpretation that nucleosomal DNA breathing substantially enhances Cas9 binding to nucleosomes and demonstrate that nucleosomal DNA sequence, through its influence on nucleosome stability, can regulate Cas9 activity over a large dynamic range.

## Nucleosome remodeling enhances Cas9 activity

We next investigated whether chromatin remodeling could enhance Cas9 activity towards chromatin substrates. Nucleosome positioning *in vivo* is strongly dependent on ATP-dependent chromatin remodelers, which are capable of loading, repositioning, and removing nucleosomes from the chromatin fiber. To measure how chromatin remodelers can influence Cas9 activity, we performed experiments where we pre-incubated 601 nucleosomes with remodeler enzymes prior to Cas9-mediated cleavage. For our experiments with the human ISWI-family remodeler SNF2h, we used asymmetric nucleosomes that possess flanking DNA only on the entry side (601 80/0 particles). When incubated with 601 80/0 particles, SNF2h promotes sliding of the nucleosome towards the center of the DNA molecule (*Figure 3A–B*, *Figure 3—figure supplement 1*) (*Längst et al., 1999*; *He et al., 2006*; *Yang et al., 2006*). We then performed *in vitro* cleavage experiments where 80/0 particles, pre-remodeled with SNF2h, were incubated with Cas9:sgRNA complex with its target site located within the nucleosome exit region. Remodeling 80/0 nucleosomes by SNF2h resulted in a strong enhancement of Cas9 cleavage to ~34%, showing that SNF2h slides the nucleosome enough to improve Cas9's accessibility to the target site and that Cas9 is now able to bind and cleave at a higher rate (*Figure 3A–D*).

We also performed this experiment by simultaneously adding SNF2h and Cas9 and found a similar rate enhancement (*Figure 3—figure supplement 2*).

While the ISWI remodeler SNF2h predominantly slides nucleosomes, remodelers from the SWI/SNF class have multiple outcomes, which include generation of DNA loops and eviction of the histone octamer in addition to nucleosome sliding (*Rowe and Narlikar, 2010*; *Narlikar et al., 2001*; *Lorch et al., 1998*; *Schnitzler et al., 1998*; *Clapier and Cairns, 2009*). To determine if the types of remodeled products generated influence Cas9 activity, we performed similar experiments using 601 80/80 particles and the yeast chromatin remodeler RSC. We find that RSC activity also dramatically

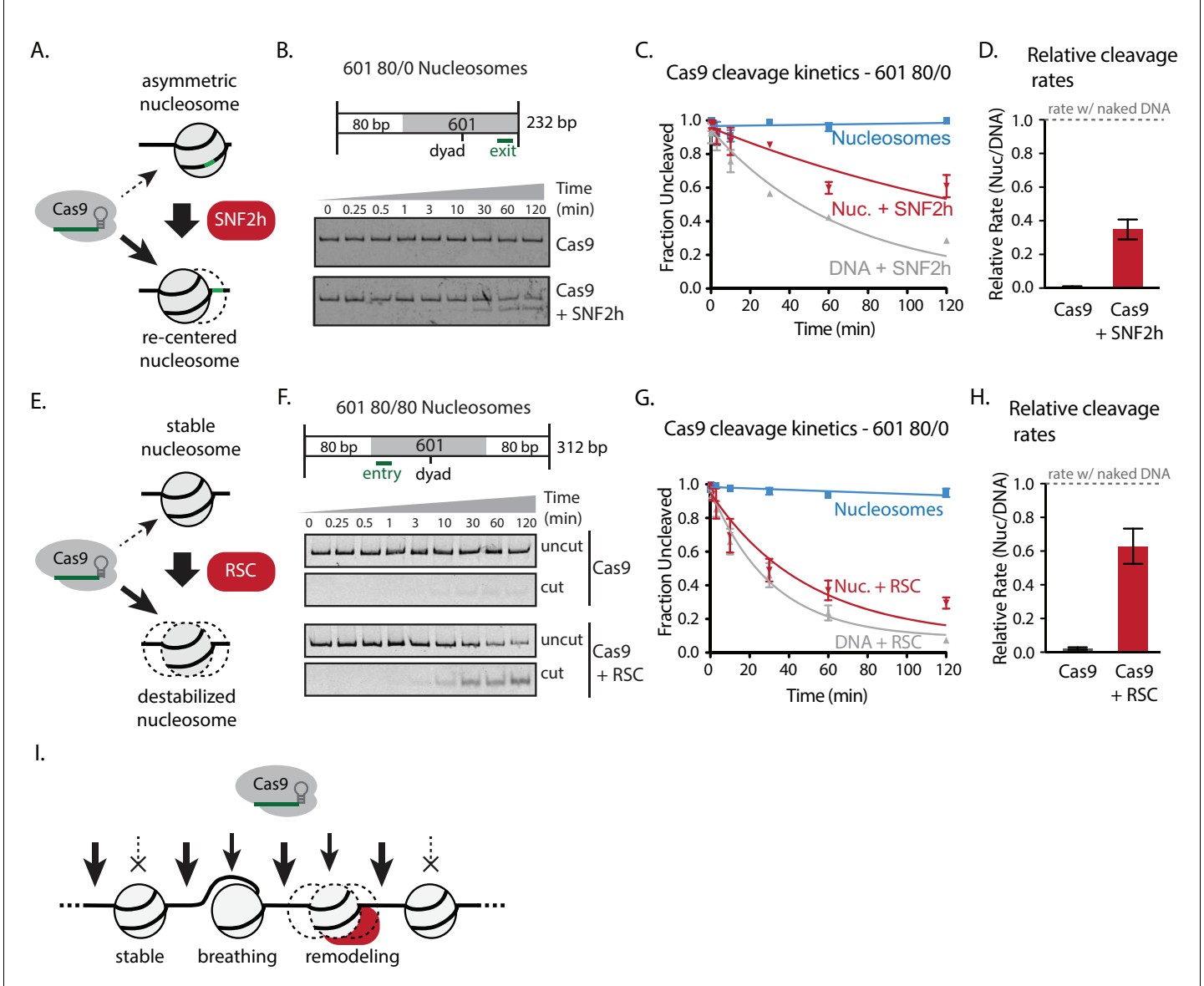

**Figure 3.** Chromatin remodeling improves Cas9 cleavage of nucleosomal substrates. (**A**) Schematic of Cas9 cleavage assay with remodeling. Cas9 is presented with 601 nucleosomes either untreated or previously remodeled with SNF2h or RSC remodelers. (**B**) Assay comparing cleavage on untreated and remodeled 80/0 nucleosomes when Cas9 is targeted to exit site (depicted in green). These asymmetric nucleosomes are recentered by SNF2h, exposing the exit target site to Cas9 (**C**) Quantification of (**B**). (**D**) Cleavage rates of 80/0 nucleosomes by Cas9 relative to naked DNA, in the presence or absence of SNF2h. SNF2h improves Cas9 cleavage to ~35% of the naked DNA cleavage rate. (**E**) Assay comparing Cas9-mediated cleavage at entry site of 80/80 symmetric 601 nucleosomes, either untreated or previously treated with RSC remodeler. RSC can destabilize nucleosome structure and reposition nucleosomes towards the DNA ends. (**F**) Quantification of (**E**) (**G**) Comparison of the rates of cleavage of nucleosomes normalized to DNA control with and without the action of RSC chromatin remodeler. Mean enhancement rates of Cas9 activity by chromatin remodeling are shown. (**H**) Cleavage rates of 80/80 nucleosomes by Cas9 relative to naked DNA, in the presence or absence of RSC. Cas9 cleavage is substantially enhanced by RSC, attaining ~63% of the naked DNA cleavage rate. Represented values are mean ± SEM from three replicates. Additional gel panels shown in *Figure 3—figure supplement 1*. (**I**) Model of Cas9 activity *in vivo* in eukaryotes. Left, stable and strongly positioned nucleosomes impede Cas9 activity (downward arrows). However, nucleosomes *in vivo* are generally more dynamic (breathing), allowing Cas9 opportunities to target underlying DNA (center). Cas9 accessibility to nucleosomal DNA can be further enhanced by the activity of chromatin remodelers that destabilize and/or reposition nucleosomes (right).

The following source data and figure supplements are available for figure 3:

**Source data 1.** Replicate gels of cleavage of 80/0 DNA and nucleosomes using sgRNA #4 with or without prior remodeling by Snf2h.

*Figure 3 continued on next page*

*Figure 3 continued*

**Source Data 2.** Replicate gels of cleavage of 80/0 DNA and nucleosomes using sgRNA #4 with or without prior remodeling by Snf2h.

**Source data 3.** Replicate gels of cleavage of 80/0 DNA and nucleosomes using sgRNA #4 with or without prior remodeling by Snf2h.

**Source data 4.** Quantification of Cas9 cleavage gels from *Figure 3—source data 1–3*.

**Source data 5.** Replicate gels of cleavage of 80/80 DNA and nucleosomes using sgRNA 601_2 with or without prior remodeling by RSC.

**Source data 6.** Replicate gels of cleavage of 80/80 DNA and nucleosomes using sgRNA 601_2 with or without prior remodeling by RSC.

**Source data 7.** Replicate gels of cleavage of 80/80 DNA and nucleosomes using sgRNA 601_2 with or without prior remodeling by RSC.

**Source data 8.** Quantification of Cas9 cleavage gels from *Figure 3—source data 5–7*.

**Figure supplement 1.** Cas9 cleavage assays with SNF2h and RSC chromatin remodelers.

**Figure supplement 2.** Simultaneous chromatin remodeling and Cas9 cleavage of nucleosomal substrates.

**Figure supplement 2—source data 1.** Gel of cleavage of 80/0 DNA and nucleosomes using sgRNA #4 with or without simultaneous remodeling by Snf2h.

**Figure supplement 3.** SNF2h and RSC remodel nucleosomes prior to Cas9 cleavage.

**Figure supplement 3—source data 1.** Test remodeling gel of 80/0 nucleosomes with Snf2h.

**Figure supplement 3—source data 2.** Test remodeling gel of 80/80 nucleosomes with RSC.

enhances cleavage on 601 80/80 nucleosomes when Cas9 is targeted to the entry site, negating most of the inhibitory influence of the nucleosome on Cas9 (*Figure 3E–F*). These results demonstrate that two different classes of chromatin remodeling enzymes can significantly enhance Cas9 access to DNA targets normally obscured by nucleosomes.

## Discussion

Here we demonstrate, using detailed biochemical studies with a variety of nucleosomal templates, that (i) the intrinsic stability of the histone-DNA interactions, (ii) the location of the target site within the nucleosomes (nucleosome positioning), and (iii) the action of chromatin remodeling enzymes play critical roles in regulating the activity of *S. pyogenes* Cas9. Below we discuss the implications of our results.

Nucleosomes have been shown to inhibit the action of DNA binding factors. Recent work using nucleosomes assembled on the 601 sequence has led to the qualitatively similar conclusion that nucleosomes are refractory for Cas9 action (*Hinz et al., 2015*; *Horlbeck et al., 2016*). The comparison here between Cas9 action on 601 nucleosomes vs. nucleosomes assembled on the native 5S sequence suggests a more refined model for how nucleosomes regulate Cas9 action. We find that Cas9 sites near the entry/exit sites of 5S nucleosomes are cleaved ~700-fold better than the corresponding sites within 601 nucleosomes. Given that DNA breathing occurs at least 100-fold more in 5S nucleosomes than 601 nucleosomes we propose that Cas9 gains access to nucleosomal DNA when the DNA is transiently unpeeled from the histone octamer. This model also explains why sites closer to the entry/exit sites are cut more readily by Cas9 than sites near the dyad. This is because DNA unpeeling up to the dyad is substantially less favored (100-fold) for both the 601 and 5S nucleosomes than DNA unpeeling near their respective entry/exit sites (*Anderson and Widom, 2000*).

*In vivo*, as *in vitro*, the precise position of nucleosomes can greatly affect DNA factor binding. Chromatin remodeling enzymes can move nucleosomes away or towards the factor binding sites to respectively enhance or inhibit factor binding. We find that Cas9 activity can also benefit from

chromatin remodeling to access nucleosomal DNA, as evidenced by the strong enhancements of Cas9 cleavage resulting from the action of the chromatin remodelers SNF2h and RSC. These two remodelers produce distinct nucleosomal arrangements yet still substantially alleviate nucleosome-mediated occlusion of Cas9 activity.

In combination, our data lead to a comprehensive model that reconciles both biochemical evidence and *in vivo* observations to explain how Cas9 is able to access nucleosomal DNA in live cells (*Figure 3I*). *In vivo*, the majority of nucleosomes are not located on strong positioning sequences, and therefore may be permissive to Cas9 binding, especially at target sites that are readily accessible by DNA unpeeling. Chromatin remodeling activities can further provide diverse mechanisms to potentiate Cas9 activity at sites located close to the nucleosomal dyad or within more strongly positioned nucleosomes, which would otherwise be refractory to Cas9 action. We hypothesize that the combination of spontaneous DNA unpeeling and remodeling contributes to the widespread success of CRISPR-Cas9 in eukaryotic cells.

Interestingly, most applications of CRISPR-Cas9 *in vivo* have focused on genome engineering of protein-coding genes and other functional genomic elements associated with gene expression, which are typically associated with high rates of nucleosome remodeling (*Clapier and Cairns, 2009*). It is also conceivable that Cas9 can temporarily gain access to less accessible regions of the genome during specific points of cell cycle (e.g. DNA replication), leading to sufficient DNA cleavage events to promote NHEJ-mediated mutagenesis or HDR-mediated DNA integration at appreciable rates. Recent studies on Cas9's behavior by single molecule imaging have also demonstrated that Cas9 favors more accessible euchromatin regions but is not completely excluded from transcriptionally silent heterochromatin (*Knight et al., 2015*). For other CRISPR applications that require stable binding of nuclease-deficient dCas9 to DNA, such as transcriptional regulation and live-cell imaging with fluorescent dCas9, even modest nucleosome phasing could have a dramatic impact (*Gilbert et al., 2013*; *Mali et al., 2013*; *Chen et al., 2013*; *Ma et al., 2015*). For example, the +1 nucleosome in RNA pol II-transcribed genes is strongly positioned with phasing that dissipates gradually with each following nucleosome. Several high resolution studies conducted in parallel to our work have established that the +1 nucleosome and resulting nucleosome phasing can exert a strong influence on dCas9's DNA-binding ability for transcriptional regulation, but the effect is less striking on genome editing with Cas9 (*Horlbeck et al., 2016*; *Smith et al., 2016*).

Our observations suggest that sgRNA design strategies that avoid targeting near the dyad of strongly phased nucleosomes are likely to be more successful than current methods. Large scale nucleosome positioning or DNA accessibility maps are now readily available and can inform CRISPR sgRNA design in order to avoid targeting regions of low accessibility (*Jiang and Pugh, 2009*; *Thurman et al., 2012*; *Wu et al., 2014*; *Hsieh et al., 2015*). Alternatively, whole cell chromatin decondensation or de-repression using chromatin factor drugs such as HDAC or DNA methyltransferase inhibitors may be an alternative and attractive strategy for improving CRISPR-Cas9 activity towards densely compact regions of chromatin (*Haaf, 1995*; *Tóth et al., 2004*).

## Materials and methods

### Cas9 and sgRNA preparation

Wild-type *Streptococcus pyogenes* Cas9 and catalytically-inactive Cas9 (dCas) containing D10A and H840A mutations were individually cloned into a custom pET-based expression vector encoding an N-terminal 6xHis-tag followed by a small ubiquitin-related modifier (SUMO) fusion tag and a Ulp1 protease cleavage site. Recombinant Cas9 variants were then expressed in *Escherichia coli* strain BL21 (DE3) (Novagen) and further purified to homogeneity as previously described (*Jiang et al., 2015*).

Single guide RNAs (sgRNAs) were prepared by *in vitro* run-off transcription using recombinant His-tagged T7 RNA polymerase and PCR product templates. Briefly, the DNA templates containing a T7 promoter, a 20-nt target DNA sequence (listed in *Table 1*) and an optimal 78-nt sgRNA scaffold were PCR amplified using Phusion Polymerase (NEB) according to manufacturer's protocol. The following PCR products were used directly as DNA templates for *in vitro* RNA synthesis in 1x transcription buffer (30 mM Tris-HCl pH 8.1, 20 mM $MgCl_2$, 2 mM spermidine, 10 mM DTT, 0.1% Triton X-100, 5 mM each NTP, and 100 µg $mL^{-1}$ T7 RNA polymerase). After incubation at 37°C for 4–8 hr, the

**Table 1.** Spacer sequences for sgRNAs used in biochemistry experiments.

| sgRNA # | Guide sequence | PAM | Target strand | Figures where used |
|---|---|---|---|---|
| 601_1 | CGAGTTCATCCCTTATGTGA | TGG | Antisense | *Figure 1D* |
| 601_2 (entry) | AATTGAGCGGCCTCGGCACC | GGG | Sense | *Figure 1D*, *Figure 2B–D*, *Figure 2—figure supplement 1*, *Figure 3E–H*, *Figure 3—figure supplement 1D–E* |
| 601_3 (core) | CCCCCGCGTTTTAACCGCCA | AGG | Antisense | *Figure 1B–D*, *Figure 1—figure supplement 1B–C*, *Figure 2B–D*, *Figure 2—figure supplement 1* |
| 601_4 | GTATATATCTGACACGTGCC | TGG | Sense | *Figure 1D* |
| 601_5 | TCGCTGTTCAATACATGCAC | AGG | Sense | *Figure 1D* |
| 601_6 | GCGACCTTGCCGGTGCCAGT | CGG | Antisense | *Figure 1D* |
| 5S_1 (entry) | TCTGATCTCTGCAGCCAAGC | AGG | Sense | *Figure 2B–E*, *Figure 2—figure supplement 1* |
| 5S_2 (core) | TATGGCCGTAGGCGAGCACA | AGG | Antisense | *Figure 2B–E*, *Figure 2—figure supplement 1* |

reactions were further treated with RNase-free DNase I (Promega) at 37°C for 30 min to remove the DNA templates. The synthesized sgRNAs were purified by Ambion MEGAclear kit and eluted into DEPC-treated $H_2O$ for downstream experiments.

## Nucleosome reconstitution

Gradient salt dialysis was used to assemble mono-nucleosomes on DNA templates containing the 147 bp long 601 or the 5S positioning sequence from *X. borealis* (listed in *Table 2*), and labeled with fluorescein on the 5' upstream end. Histones and histone octamers were prepared as previously described (*Luger et al., 1999*).

## Cas9 cleavage assays

Cleavage assays were conducted as previously described with the following modifications (*Anders and Jinek, 2014*). Cas9:sgRNA complexes were reconstituted by incubating Cas9 and sgRNA for 10 min at 37°C. Reactions contained 5 nM fluorescein labeled DNA or nucleosomes and 100 nM Cas9:sgRNA. In combined cleavage and remodeling experiments, 25 nM SNF2h or 3 nM RSC was first incubated with 5 nM naked DNA or nucleosomes for 45 min at 37°C (*Narlikar et al., 2001*). Cleavage assays were carried out in reaction buffer (20 mM Tris-HCl pH 7.5, 70 mM KCl, 5 mM $MgCl_2$, 5% Glycerol, and 1 mM DTT) at 25°C. For SNF2h and RSC remodeling experiments, 0.2 mM ATP was also added. For RSC remodeling experiments, 1 mM $MgCl_2$ was used. Time points were quenched using stop buffer (20 mM Tris-HCl pH 7.5, 70 mM EDTA, 2% SDS, 20% glycerol, and 0.2 mg/mL xylene cyanol and bromophenol blue). Proteins were digested with 3 mg/mL of Proteinase K and incubated at 50°C for 20 min. Samples were resolved on 1x TBE, 10% Polyacrylamide gels

**Table 2.** Sequences for DNA molecules used for biochemical assays (Positioning sequence highlighted in grey).

| Name | Sequence |
|---|---|
| 601 80/80 | CGGGATCCTAATGACCAAGGAAAGCATGATTCTTCACACCGAGTTCATCCCTTATGTGATGGACCCTATACGCGGCCGC CCTGGAGAATCCCGGTGCCGagGCCGCTCAATTGGTCGTAGACAGCTCTAGCACCGCTTAAACGCACGTACGCGCTGTCCCCC CGCGTTTTAACCGCCAAGGGGATTACTCCCTAGTCTCCAGGCACGTGTCAGATATATACATCCTGTGCATGTATTGAAC AGCGACCTTGCCGGTGCCAGTCGGATAGTGTTCCGAGCTCCCACTCTAGAGGATCCCCGGGTACCGA |
| 601 0/0 | CTGGAGAATCCCGGTGCCGagGCCGCTCAATTGGTCGTAGACAGCTCTAGCACCGCTTAAACGCACGTACGCGCTGTCCCCC GCGTTTTAACCGCCAAGGGGATTACTCCCTAGTCTCCAGGCACGTGTCAGATATATACATCCTGT |
| 601 80/0 | CGGGATCCTAATGACCAAGGAAAGCATGATTCTTCACACCGAGTTCATCCCTTATGTGATGGACCCTATACGCGGCCGC CCTGGAGAATCCCGGTGCCGagGCCGCTCAATTGGTCGTAGACAGCTCTAGCACCGCTTAAACGCACGTACGCGCTGTCCCCC CGCGTTTTAACCGCCAAGGGGATTACTCCCTAGTCTCCAGGCACGTGTCAGATATATACATCCTGT |
| 5S 0/0 | GGCCCGACCCTGCTTGGCTGCAGAGATCAGACGATATCGGGCACTTTCAGGGTGGTATGGCCGTAGGCGAGCACAAGGCT GACTTTTCCTCCCCTTGTGCTGCCTTCTGGGGGGGGGCCCAGCCGGATCCCCGGGCGAGCTCGAATT |

for 4 hr at 140 V before visualizing using a Typhoon scanner (GE Healthcare) and quantifying with Image J (*Schneider et al., 2012*). For band quantification, background intensity was first subtracted after averaging the intensity of three areas. For cleavage gels, fraction uncleaved was determined by measuring the intensity of the uncleaved band compared to the total intensity for the lane. Similarly, fraction unbound was determined by measuring the intensity of the unbound band compared to the total intensity for the lane.

All experiments were performed in triplicate. Experiment variability is presented as the standard error of the mean, calculated by the standard deviation divided by the square root of N.

Propagation of error for Rates of Cleavage on Nucleosomes to Rates of Cleavage on DNA was calculated as follows:

$$Error = \frac{k_{Nucleosome}}{k_{DNA}} \sqrt{\left(\frac{SEM_{Nucleosomes}}{k_{Nucleosomes}}\right)^2 + \left(\frac{SEM_{DNA}}{k_{DNA}}\right)^2}$$

Data were fit on Graphpad Prism using a standard one phase decay model:

$$Y = (Y_0 - Plateau)e^{-kt} + Plateau$$

where Y is the fraction of uncleaved DNA, $Y_0$ is the value of Y at time = 0, k is the observed rate constant (min$^{-1}$) and t is time (min).

### Native gel mobility shift assays

dCas9 and a 2x molar ratio of sgRNA were incubated for 10 min at 37°C. Various concentrations of dCas9:sgRNA complex were incubated with 20 nM naked DNA or nucleosomes in binding buffer (20 mM Tris-HCl pH 7.5, 100 mM KCl, 5 mM MgCl$_2$, 5% Glycerol, 1 mM DTT, and 0.02% NP-40). Samples were incubated at room temperature for 1 hr before being run on native 0.5X TBE 6% polyacrylamide gels, visualized on a Typhoon scanner, and quantified using ImageJ. Fraction unbound was measured as the intensity of all unbound species divided by the total intensity. Fraction unbound was then converted to fraction bound:

$$Fraction\,Bound = 1 - Fraction\,Unbound,$$

and binding curves were fit with:

$$Fraction\,Bound = \frac{[Cas9:sgRNA]^n}{([Cas9:sgRNA]^n + K_{1/2}^n)}$$

## Acknowledgements

We would like to thank members of the Narlikar Lab, especially Nathan Gamarra, Coral Zhou, Kalyan Sinha, and Stephanie Johnson for providing reagents and assistance and members of the Lim lab, especially Scott Coyle, Levi Rupp, Amir Mitchell and Russell Gordley for assistance and helpful discussions during the planning and preparation of this manuscript.

## Additional information

### Competing interests

JAD: Co-founder of Caribou Biosciences; Editas Medicine; Intellia Therapeutics. WAL: Founder of Cell Design Labs, and member of its scientific advisory board. The other authors declare that no competing interests exist.

### Funding

| Funder | Grant reference number | Author |
|---|---|---|
| Merck Fellow of the Damon Runyon Cancer Research Foundation | DRG-2201-14 | Fuguo Jiang |

| National Science Foundation | 1244557 | Jennifer A Doudna |
|---|---|---|
| National Institutes of Health | R01 DA036858 | Wendell A Lim |
| National Institutes of Health | P50 GM081879 | Wendell A Lim |
| National Institutes of Health | R01 GM073767 | Geeta J Narlikar |
| Howard Hughes Medical Institute | | Jennifer A Doudna Wendell A Lim |

The funders had no role in study design, data collection and interpretation, or the decision to submit the work for publication.

## Author contributions

RSI, Conceived of and conducted the biochemistry experiments and data analysis, Helped write this report; FJ, Generated reagents used in experiments, Edited this report; JAD, Contributed ideas and reagents, Edited this report; WAL, GJN, Co-supervised the work, Helped write this report; RA, Conceived of and conducted the work, Generated reagents, Wrote this report

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
