## [Decision Letter]

[Editors’ note: this article was originally rejected after discussions between the reviewers, but the paper was accepted for publication after an appeal against the decision.]

Thank you for submitting your work entitled "Nucleosome Positioning, Dynamics and Remodeling Constrain CRISPR-Cas9 Function" for consideration by *eLife*. Your article has been reviewed by two peer reviewers, one of whom is a member of our Board of Reviewing Editors, and Jessica Tyler as the Senior Editor. Our decision has been reached after consultation between the reviewers.

Based on these discussions and the individual reviews below, we regret to inform you that your work will not be considered further for publication in *eLife*. The reviewers agree that the work was performed to a high standard and addresses an important problem. However, the advances represented by this study are relatively small in light of recent publications on this topic. Although the reviewers did have suggestions for increasing impact, we feel that these fall outside the realm of typical revisions at *eLife*.

Reviewer #1:

This short article from the Narlikar lab presents a detailed biochemical analysis of the effects of nucleosomes on binding and activity of CRISPR-Cas9. The main take-home is that nucleosomes block binding of Cas9 to PAM sites, thereby strongly inhibiting CRISPR-Cas9 cleavage activity.

The authors begin by studying cleavage of mononucleosomal DNA comprising the 601 element with 80 bp of DNA on either side. They use sgRNAs that target sites within the nucleosome dyad, at the entry/exit sites, or within linker DNA. In full agreement with recently published work (Hinz et al. Biochemistry 2015), they find that linker DNA is efficiently cleaved, but that DNA sites near the dyad are nearly completely protected by the nucleosome. Presumably due to DNA breathing, a low level of cleavage is observed at entry/exit sites. This data is clear and very well presented.

Next, the authors investigate which step is inhibited, and use gel shifts to demonstrate that the initial binding of Cas9 to PAM-containing DNA is blocked by nucleosomes. However, they also report a curious, non-specific binding of Cas9 to linker DNA that obscures this result, and forces them to use only 601 particles lacking linker DNA. This detracts from the resulting conclusions in that they can't compare specific binding near the dyad to binding at entry sites or linker regions. But the basic finding remains convincing: chromatin blocks Cas9 from binding PAM sites.

Finally, the authors show that if a chromatin remodeler is used to move the nucleosome off the PAM site being targeted before Cas9 is mixed with DNA, then it can be more efficiently cleaved. I am not clear on what more we learn from this experiment, other than that it's another way to show that the position of a PAM site within a nucleosome dictates the efficiency of cleavage by CRISPR-Cas9.

Overall, the manuscript is well written, the experiments are very nicely performed and presented, and the findings are compelling. However, I am not sure how much new is learned here beyond previous publications. The position-specific inhibition of CRISPR-Cas9 cleavage by nucleosomes was recently shown using nearly identical assays. So the novel aspect of this work is the delineation of the binding step as the point of inhibition, rather than unwinding or later steps. This is nice, but is I am not sure whether it represents a big step forward in our understanding, thus my enthusiasm for this work is only moderate. I wonder if the authors could perhaps provide more new information by performing an analysis of cleavage during remodeling (rather than afterwards) and by using a remodeler that doesn't reposition nucleosomes, but just makes them more dynamic, generating loops of potentially accessible DNA, etc.

Also, all the work thus far has been done on the 601 DNA sequence, which is much more stable and translationally inflexible than other sequences, and this must be considered in interpretation.

Reviewer #2:

This manuscript studies the effect of nucleosome position on Cas9 binding to and cleavage of DNA targets. This is a very important subject given the current poor understanding of how Cas9 interacts with chromatinized DNA. This is a short study simply showing reduced cutting by Cas9 when nucleosomes are present. This is an interesting observation, but the cursory nature of the investigation and the previous report of essentially identical results decreases the overall impact of the work. The study would be significantly strengthened by a deeper analysis how Cas9 finds and interacts targets on nucleosomal and non-nucleosomal DNA.

1) The 2015 paper from Hinz, Laughery and Wyrick shows essentially identical results to the main points of this study. Although the current study includes the analysis of Cas9 cutting in the present of Snf2h, this is not assessed in depth.

2) Only a single gRNA and target site combination is studied. Because of the widely observed differences in activity of different gRNAs, it is important to study different gRNAs and determine that the results shown here are not unique to one or more particular targets. A similar problem is shown in Figure 1, where the experiment is designed to study the variable of position within the nucleosome, but the variable of sequence identity of the target site is also changing and therefore convoluting the results. Since only a single target DNA strand was analyzed, it's difficult to generalize the observations. It would strengthen the conclusions to evaluate target site position independent of sequence.

3) The authors should consider making an additional effort to resolving their model with the many published studies showing Cas9 gene editing and dCas9 binding in heterochromatin. Some of these studies even show remodeling of chromatin in mammalian cells – how might this happen given the new results shown here? Does the proposed model in Figure 3 help predict gene editing activity within eukaryotic cells? Or does it at least explain differences in gene editing efficiency between target sites already described in the literature? Can the authors use the referenced nucleosome positioning maps to explain published data (or new data on gene editing in cells that they generate)?

---

## [Author Response]

[Editors’ note: the author responses to the first round of peer review follow.]

We are writing to appeal the decision by *eLife* on our manuscript, which was being considered as a co-submission with the manuscript by Weissman and colleagues. The critiques by the reviewers were very helpful in focusing us on the key issues of understanding how nucleosomes impede Cas9. Thus, in the last few weeks we have gathered significant additional data that addresses the core concerns of the reviewers and substantially widens the scope of our conclusions. The data provides a far deeper understanding of Cas9-nucleosome interactions.

Our new results challenge the emerging simple view that nucleosomes are inhibitory. We find that nucleosomes assembled on native DNA sequences readily allow Cas9 action at locations that are inhibited when artificially derived nucleosome positioning sequences are used (the 601 sequence that has been used in all the published studies to date). These data uncover the dynamic range of Cas9 activity as influenced by nucleosomes and provide a mechanistic framework to understand how nucleosome dynamics can be leveraged to enhance Cas9 action *in vivo*.

The reviewers agree that the work was performed to a high standard and addresses an important problem. However, the advances represented by this study are relatively small in light of recent publications on this topic. Although the reviewers did have suggestions for increasing impact, we feel that these fall outside the realm of typical revisions at eLife. eLife is highly selective, which means that the majority of submissions are rejected, but we thank you for sending your work for review and we hope you will submit to eLife again in the future.

Both reviewers thought the paper was well written and technically compelling, but their major criticism was that our result that nucleosomes can block Cas9 cleavage has been previously shown. In response to these criticisms, we have significantly improved the paper by adding several critical pieces of new data that provide a far more nuanced and refined view of the important question of how Cas9 can interact with eukaryotic chromatinized DNA. We thank the reviewers for their excellent suggestions.

These new data are:

1) We have examined Cas9 cleavage of more natural nucleosome particles. All prior studies (including our original data) have been performed using the 601 sequence element, which is an in vitro selected sequence that positions nucleosomes more rigidly compared to most native sequences. We have now repeated these studies with the naturally derived 5S element, which is one of the strongest natural sequences with respect to nucleosome positioning. Strikingly we find that the 5S DNA is up to 700-times more accessible to Cas9 than the artificial 601, although the dyad is still well protected. These data suggest a revised model – that many natural nucleosomes have sufficient dynamic breathing properties to allow for Cas9 interrogation (except for regions close to the dyad).

2) We have tested the effects of a second chromatin remodeler (RSC) that functions by a different mechanism than SNF2h. RSC slides nucleosomes and, additionally, generates DNA loops. We now find that RSC has even greater ability than SNF2h to improve Cas9 accessibility.

Thus our new results provide a newer, more refined model of how Cas9 can interact with and interrogate chromatinized DNA. Our findings suggest that natural nucleosomes might have sufficient dynamic DNA breathing to allow for Cas9 searching and binding. Moreover, several classes of remodelers that reposition or destabilize nucleosomes can also favor Cas9 binding. This more comprehensive picture of Cas9 interactions with nucleosomes provides an explanation for why Cas9 has empirically proven so successful in accessing many endogenous loci. This model also provides ideas for how one can optimally pick Cas9 targets or how accessibility could be improved. Overall, we feel that this paper now provides significant advances over the prior work on this subject.

Reviewer #1:

*This short article from the Narlikar lab presents a detailed biochemical analysis of the effects of nucleosomes on binding and activity of CRISPR-Cas9. The main take-home is that nucleosomes block binding of Cas9 to PAM sites, thereby strongly inhibiting CRISPR-Cas9 cleavage activity.*

The authors begin by studying cleavage of mononucleosomal DNA comprising the 601 element with 80 bp of DNA on either side. They use sgRNAs that target sites within the nucleosome dyad, at the entry/exit sites, or within linker DNA. In full agreement with recently published work (Hinz et al. Biochemistry 2015), they find that linker DNA is efficiently cleaved, but that DNA sites near the dyad are nearly completely protected by the nucleosome. Presumably due to DNA breathing, a low level of cleavage is observed at entry/exit sites. This data is clear and very well presented.

Next, the authors investigate which step is inhibited, and use gel shifts to demonstrate that the initial binding of Cas9 to PAM-containing DNA is blocked by nucleosomes. However, they also report a curious, non-specific binding of Cas9 to linker DNA that obscures this result, and forces them to use only 601 particles lacking linker DNA. This detracts from the resulting conclusions in that they can't compare specific binding near the dyad to binding at entry sites or linker regions. But the basic finding remains convincing: chromatin blocks Cas9 from binding PAM sites.

Finally, the authors show that if a chromatin remodeler is used to move the nucleosome off the PAM site being targeted before Cas9 is mixed with DNA, then it can be more efficiently cleaved. I am not clear on what more we learn from this experiment, other than that it's another way to show that the position of a PAM site within a nucleosome dictates the efficiency of cleavage by CRISPR-Cas9.

*Overall, the manuscript is well written, the experiments are very nicely performed and presented, and the findings are compelling. However, I am not sure how much new is learned here beyond previous publications. The position-specific inhibition of CRISPR-Cas9 cleavage by nucleosomes was recently shown using nearly identical assays. So the novel aspect of this work is the delineation of the binding step as the point of inhibition, rather than unwinding or later steps. This is nice, but is I am not sure whether it represents a big step forward in our understanding, thus my enthusiasm for this work is only moderate.*

We thank the reviewer for their careful reading of the manuscript. We have used the suggestions made by the reviewer to strengthen the study and increase its impact as described below in a point-by-point manner.

I wonder if the authors could perhaps provide more new information by performing an analysis of cleavage during remodeling (rather than afterwards) and by using a remodeler that doesn't reposition nucleosomes, but just makes them more dynamic, generating loops of potentially accessible DNA, etc.

We have now performed the experiment with analysis of cleavage during remodelling by SNF2h and we observe similar rate enhancements. Simultaneous cleavage results in a rate that is ~40% compared to naked DNA while sequential cleavage increases to ~35% compared to naked DNA (see Figure 3—figure supplement 2). In terms of the reviewer’s latter suggestion we have now investigated the effects of another chromatin remodeler, RSC. In contrast to SNF2h, which only slides nucleosomes away from DNA ends, RSC generates a wider range of remodelled products that are thought to include nucleosomes with DNA loops in addition to nucleosomes with different translational positions. We find that RSC activity also enhances Cas9 cleavage and has a larger stimulatory effect than SNF2h. These results strengthen our model that chromatin remodeling activities can potentiate Cas9 activity on nucleosomal DNA *in vivo*.

Also, all the work thus far has been done on the 601 DNA sequence, which is much more stable and translationally inflexible than other sequences, and this must be considered in interpretation.

The reviewer makes an excellent point. The published Biochemistry study as well as our first submission used the artificial 601 sequence to assemble nucleosome core particles. Previous work by the Widom group has shown that nucleosomes assembled on the 601 sequence show ~100-fold less DNA breathing than the naturally occurring 5S sequence (Anderson et al. J. Mol. Biol., 2000). To test if such increased DNA breathing increases Cas9 activity we compared Cas9 cutting on 5S nucleosome particles. With these more physiological 5S nucleosomes we detect ~700-fold higher Cas9 cleavage rates compared to rates at the same relative position on 601 nucleosomes. Our new results (i) strengthen our previous model that DNA breathing dynamics regulate Cas9 activity towards nucleosomal DNA and (ii) demonstrate that nucleosomal DNA sequence can regulate Cas9 activity over a large dynamic range. The results also imply that most nucleosomes *in vivo*, which are located on weaker positioning sequences than 5S, may not present large impedances to Cas9.

Reviewer #2:

*This manuscript studies the effect of nucleosome position on Cas9 binding to and cleavage of DNA targets. This is a very important subject given the current poor understanding of how Cas9 interacts with chromatinized DNA. This is a short study simply showing reduced cutting by Cas9 when nucleosomes are present. This is an interesting observation, but the cursory nature of the investigation and the previous report of essentially identical results decreases the overall impact of the work. The study would be significantly strengthened by a deeper analysis how Cas9 finds and interacts targets on nucleosomal and non-nucleosomal DNA.*

We thank the reviewer for their suggestions. We have incorporated new data that helps address the core concerns as detailed below.

*1) The 2015 paper from Hinz, Laughery and Wyrick shows essentially identical results to the main points of this study. Although the current study includes the analysis of Cas9 cutting in the present of Snf2h, this is not assessed in depth.*

As the reviewer mentioned, our results do indeed confirm the observations recently reported in Hintz et al. Biochemistry 2015, but our initial study built on the published work by (a) showing that nucleosomes inhibited the binding step of Cas9 to PAMs and (b) that remodelling enzymes can alleviate the inhibitory effect of a nucleosome. However, both the reviewers’ comments led us to perform additional experiments to probe more deeply how nucleosomes regulate Cas9 activity.

a) We find that changing the DNA sequence from the strong and artificial positioning sequence, 601, to a naturally occurring positioning sequence, 5S, dramatically increases (by ~700-fold) the ability of Cas9 to access nucleosomal DNA. This result is consistent with old observations that 5S nucleosomal DNA breathes ~100-fold more than 601 nucleosomal DNA (Anderson et al., J. Mol Biol., 2000). Our new results thus (i) strengthen our previous model that DNA breathing dynamics regulate Cas9 activity towards nucleosomal DNA and (ii) demonstrate that nucleosomal DNA sequence can regulate Cas9 activity over a large dynamic range. In comparison, the published Biochemistry study used solely 601 nucleosomes.

b) We have now investigated the effects of another chromatin remodeler, RSC. In contrast to SNF2h, which only slides nucleosomes away from DNA ends, RSC generates a wider range of remodeled products that are thought to include nucleosomes with DNA loops in addition to nucleosomes with different translational positions. We find that RSC activity also enhances Cas9 cleavage and has a larger stimulatory effect than SNF2h, bringing the rate of cutting to nearly the same level as on naked DNA.

Together, the two new results outlined in (a) and (b) lead to the following, more comprehensive model to explain how Cas9 is able to access nucleosomal DNA i*n vivo*: Most nucleosomes *in vivo*, are located on weaker positioning sequences than 5S, and thus may not present large intrinsic impedances to Cas9, especially when the sites are accessible through DNA breathing. For Cas9 targets that are either present closer to the dyad of the nucleosome or present within nucleosomes occupying strong positioning sequences, chromatin remodeling activities can provide diverse mechanisms to potentiate Cas9 activity.

2) Only a single gRNA and target site combination is studied. Because of the widely observed differences in activity of different gRNAs, it is important to study different gRNAs and determine that the results shown here are not unique to one or more particular targets. A similar problem is shown in Figure 1, where the experiment is designed to study the variable of position within the nucleosome, but the variable of sequence identity of the target site is also changing and therefore convoluting the results. Since only a single target DNA strand was analyzed, it's difficult to generalize the observations. It would strengthen the conclusions to evaluate target site position independent of sequence.

The reviewer raises a critical point. To control for the intrinsic variations introduced by the different guide RNA sequences we always normalize nucleosome cleavage rates for each individual sgRNA to their rates on the naked DNA target (Figure 1). This normalization isolates nucleosome specific effects on Cas9 activity, allowing comparisons across multiple nucleosome positions. We have better clarified the normalization procedure in the revised manuscript. Furthermore, the activity of the sgRNAs used in our experiments was confirmed using cleavage assays with naked DNA. sgRNA designs found to be substantially less effective were replaced with alternative designs with similar target sites in order to preserve the dynamic range of our experiments.

3) The authors should consider making an additional effort to resolving their model with the many published studies showing Cas9 gene editing and dCas9 binding in heterochromatin. Some of these studies even show remodeling of chromatin in mammalian cells – how might this happen given the new results shown here? Does the proposed model in Figure 3 help predict gene editing activity within eukaryotic cells? Or does it at least explain differences in gene editing efficiency between target sites already described in the literature? Can the authors use the referenced nucleosome positioning maps to explain published data (or new data on gene editing in cells that they generate)?

We agree with the reviewer that reconciling our observations that the strong effect nucleosomes have on Cas9 activity with widespread observations of Cas9 activity *in vivo* is of great importance. We address these issues below.

i) As described in response to the Major concern #1 above, the additional data in this revision leads to a comprehensive model in which, (a) Cas9 acts on nucleosomes occupying weak positioning sequences without substantial inhibition and (b) Cas9 opportunistically benefits from the action of different remodelers to act on nucleosomes assembled on strong positioning sequences.

ii) As the reviewer points out dCas9 binding *in vivo* can increase DNA accessibility in regions that are occupied by nucleosomes. Our results indicate that Cas9 can readily act at the entry/ exit sites of nucleosomes assembled on native DNA sequences like 5S by taking advantage of DNA breathing. We speculate that when dCas9 accesses such sites, its slow off rate keeps the unpeeled DNA from re-annealing with the histone octamer, thus making the histones more susceptible to disassembly by cellular histone chaperones or chromatin remodelers.

iii) The reviewer suggests performing a correlation analysis on nucleosome position maps and published gene editing efficiency datasets to validate our mechanistic model. Previous *in vivo* dCas9 binding studies have already established that nucleosome-free regions are more amenable to Cas9 binding (Kuchu et al. and Wu et al., Nature Biotechnology, 2014). Furthermore, a higher resolution study on Cas9 editing/gene regulation efficiencies *in vivo* using purposefully designed, high density sgRNA libraries over a large number of genes is described in a parallel studies in human cells (Horlbeck and Witkowsky et al., *eLife*, 2016) and in yeast (Smith et al., Genome Biology, 2016) where the authors find that Cas9 function is negatively correlated with nucleosome positioning. These *in vivo* findings are in agreement with, and complementary to our mechanistic biochemical studies.